# Cannabinoid Regulation of Murine Vaginal Secretion

**DOI:** 10.3390/biom15040472

**Published:** 2025-03-24

**Authors:** Natalia Murataeva, Sam Mattox, Kyle Yust, Alex Straiker

**Affiliations:** 1Department of Psychological and Brain Sciences, Indiana University, Bloomington, IN 47401, USA; nmuratae@iu.edu (N.M.); kyleay527@gmail.com (K.Y.); 2Gill Institute for Neuroscience, Indiana University, Bloomington, IN 47405, USA

**Keywords:** vaginal, cannabinoid, vaginal dryness

## Abstract

Tearing and salivation are wholly dependent on the activity of exocrine (lacrimal and salivary) glands, whereas vaginal moisture and secretion rely on a combination of exudation and exocrine secretion. Exocrine gland disorders impact millions, and women with Sjögren’s Syndrome often experience dry eye and mouth as well as vaginal dryness. Cannabis users’ complaints of dry eye and ‘cottonmouth’ are well-known, but some female cannabis users also report vaginal dryness. The regulation of vaginal secretion by the cannabinoid signaling system is essentially unstudied. We recently reported that despite their small size and nocturnal nature, laboratory mice have measurable basal vaginal moisture and pheromone-stimulated secretory responses that are regulated by circadian and estrous factors. We tested the regulation of vaginal moisture by cannabinoid CB1 receptors in this model. We now report that the cannabinoid receptor agonist CP55940 does not alter baseline vaginal moisture but prevents a stimulated secretory response due to a local peri-vaginal effect. Chronic intermittent CP55940 reduces basal vaginal moisture but also unmasks or induces a potentiating effect for CP55940, suggesting multiple sites of action. The acute and chronic effects likely occur via CB1 receptors. Δ^9^-tetrahydrocannabinol (THC), the chief psychoactive ingredient of cannabis, a partial agonist at CB1, has no acute or chronic effects. In summary, strong acute activation of CB1 receptors in a murine model does not reduce vaginal moisture but does prevent a pheromone-stimulated vaginal secretory response. In contrast, chronic intermittent CB1 activation reduces baseline vaginal moisture. The extent to which these findings translate to humans remains to be determined.

## 1. Introduction

Cannabis users famously complain of dry mouth and dry eye (e.g., [1]), but there are also informal reports of vaginal dryness from some female cannabis users. Tearing and salivation are wholly dependent on the activity of exocrine (lacrimal and salivary) glands. In humans, vaginal moisture derives from multiple sources. Basal moisture is largely due to transudate diffusion across the capillary membranes in the vaginal wall [2]. Arousal-related secretions are thought to derive from Bartholin’s glands, mucinous exocrine glands under control of the pudendal nerve that lubricate the vaginal opening and surrounding vulva [3,4]. The paraurethral/Skene’s glands [5] are also implicated in vaginal lubrication [6]. Exocrine gland disorders impact millions, especially women and the elderly, in the form of Sjögren’s Syndrome, an inflammatory disease of exocrine glands that results in dry eye and mouth, as well as vaginal dryness [7].

The regulation of vaginal secretion by the cannabinoid signaling system has seen limited study though a survey of cannabis users’ perception of their sexual function reported no significant change in vaginal lubrication [8]. We have examined the cannabinoid regulation of lacrimal [9] as well as both submandibular [10] and parotid [11] glands. These glands have a similar cannabinoid architecture with presynaptic CB1 receptors on parasympathetic inputs. The exception thus far is female lacrimal glands, with an additional population of CB1 receptors on myoepithelial cells that may contribute to circadian regulation of tearing [12]. Is it possible that cannabinoid receptors similarly regulate the exocrine component of vaginal secretion?

The cannabinoid signaling system is still best known as the target of the chief psychoactive ingredient of cannabis, Δ^9^-tetrahydrocannabinol (THC), but this signaling system regulates many important physiological processes. The canonical cannabinoid signaling system includes two G protein-coupled receptors, CB1 and CB2 [13,14], that together are found throughout the body, though CB1 and CB2 appear to be especially abundant in the nervous and immune systems, respectively [15,16]. These receptors are activated by endogenous lipid messengers that are produced ‘on demand’ and then metabolized enzymatically [17]. THC-based therapies have been used to treat nausea and to promote appetite [18]. Pharmacological effects on vaginal moisture and secretion cannot be easily studied in female humans; hence the actual effect of cannabinoids on vaginal function is essentially unstudied, partly for want of animal models. We recently reported that despite their small size and nocturnal nature, laboratory mice have measurable vaginal moisture and stimulated secretory responses, and so may serve as a model for the study of vaginal secretory function in mammals [19]. Using this model, we have examined the regulation of vaginal moisture and secretion by the cannabinoid signaling system.

## 2. Methods

### 2.1. Study Animals

Mice were group-housed in specific-pathogen-free conditions in standard ventilated caging (cage floor dimensions: 33.5 × 18 cm, depth 14 cm), with corn cob-based bedding (Bed-o’combs laboratory animal bedding, The Andersons, Maumee, OH, USA). Mice were group-housed 3–4 mice per cage and were fed Inotiv Teklad 2918 irradiated rodent diet (Inotiv, Lafayetteville, IN, USA) ad libitum. Because mice are nocturnal, we tested their responses during their active phase by maintaining them on a ‘reverse’ light cycle.

As a rule, CD1 strain mice were used. CD1 strain mice were chosen in part because the CB1 receptor knockout mice are maintained on this background strain but also because we have found that, in contrast to C57BL/6 mice, they have a clear circadian tearing response [12]. Wild type and CB1 knockout CD1 strain mice were bred in a colony in the same facility and kindly provided by Dr. Ken Mackie (Indiana University, Bloomington, IN, USA). The CB1 knockout mice were originally provided as heterozygotes from the laboratory of Dr. Catherine Ledent (Université Libre de Bruxelles, Brussels, Belgium) [20]. Mice were 2–6 months of age. All animal care and experimental procedures used in this study were approved by the Institutional Animal Care and Use Committee of Indiana University and conform to the Guidelines of the National Institutes of Health on the Care and Use of Laboratory Animals. Experiments complied with ARRIVE guidelines.

### 2.2. Experimental Design

#### 2.2.1. Sex and Disposition of Animals

Only adult female mice were tested for vaginal responses, but bedding samples were obtained from males. Mice were housed in the same room in the mouse colony, but the males and females were sexually naïve, meaning that the females had not had an opportunity to physically interact with males. A given cage of CD1 strain females were siblings.

#### 2.2.2. Number of Mice Used for Each Experiment

Treatments were via Intraperitoneal (IP) injection or Intravaginal (IVag) delivery via pipette. Drugs employed were the cannabinoid receptor agonist CP55940 (CP), CB1-selective antagonist SR141716 (SR), or the psychoactive cannabis constituent tetrahydrocannabinol (THC). Dose determinations are described under the subheading Drugs below.

Acute treatments

(1)THC, IP 4 mg/kg, active phase, *n* = 16;(2)THC, IP 10 mg/kg, active phase, *n* = 8;(3)THC, IP 4 mg/kg, rest phase, *n* = 8;(4)THC, Intravaginal (IVag) 1 mM (10 μL), *n* = 9;(5)CP55940, IP 0.5 mg/kg, *n* = 10;(6)CP55940, IntraVag 150 μM (10 μL), *n* = 13;(7)SR141716, IP 4 mg/kg, *n* = 12.

Stimulated responses with control vs. cannabinoid receptor agonist CP55940

(1)CP55940, IP 0.5 mg/kg, *n* = 12;(2)Vehicle, IP, *n* = 13;(3)CP55940, IVag 150 μM (10 μL), *n* = 30.

Stimulated responses in CB1 knockouts and after CB1 receptor antagonist SR141716

(1)CB1 KO vs. WT (CD1 strain) baselines, *n* = 30, 32;(2)CB1 KO bedding responses, *n* = 16;(3)WT, SR141716, IP 4 mg/kg, *n* = 24;(4)WT, SR141716, IVag 1 mM (10 μL), *n* = 15.

Chronic THC experiments

(1)THC, IP 4 mg/kg, daily, *n* = 10;(2)THC, IP 4 mg/kg, chronic (weekly), *n* = 16.

Chronic CP55940 experiments

(1)CP55940, IP 0.5 mg/kg (daily), CD1 strain, *n* = 13;(2)CP55940, IP 0.5 mg/kg (weekly), CD1 strain, *n* = 7;(3)CP55940, IP 0.5 mg/kg (weekly), CB1 knockout on CD1 strain, *n* = 8.

#### 2.2.3. Measurement of Vaginal Moisture

The use of colorimetric threads for the measurement of vaginal moisture as well as the method of manufacture of these threads was described recently [19]. Briefly, to allow for consistent placement in the vaginal cavity, a phenol red-infused colorimetric thread is inserted into a glass capillary (cat#:593200, AM Systems, Sequim, WA, USA; 0.86 mm inner diameter, fire-polished to prevent injury), leaving 3 mm of thread outside the opening of the capillary. The capillary is then placed into the vaginal cavity of an unanesthetized mouse for 10 s. The distance traveled along the thread is quantified and taken as a measure of vaginal moisture.

In the case of intravaginal treatment, drugs were dissolved in 10% (2-hydroxypropyl) β cyclodextrin in PBS (pH adjusted to 7.0 with NaOH) and administered by pipette (10 μL) into the vaginal cavity. The experimental timeline was to obtain a baseline measurement of vaginal moisture, administer the drug or vehicle intravaginally, and then re-test vaginal moisture after an hour (i.e., an hour passed between intravaginal treatment and subsequent measurement). No change in baseline moisture was seen with vehicle.

### 2.3. Bedding

Bedding for a given experiment was obtained in coordination with the weekly animal facility cage change. Bedding (from 6 days of habitation) was obtained from a cage of 3–4 male mice. Bedding was used within 1 h. For bedding exposure, males were relocated to a clean cage, and females were moved into the soiled male cage and allowed to roam for 1 h, after which the vaginal moisture measurement was obtained, and females were moved back into their home cage. A given soiled male cage was used only once (for one cage of females).

#### 2.3.1. General Method of Exposure to Odorant (Bedding)

For odorant exposure experiments, several cages containing a group of CD1 strain female mice (3–4/cage) were brought into an adjoining room separated by a closed door. Mice were tested in the late morning, starting at 10 AM, to reduce the likelihood of unusual scents. All baseline vaginal moisture readings were obtained for each mouse as described under Measurement of Vaginal Moisture above. This took less than a minute per mouse, so it was accomplished in under half an hour. After this, the first cage of female mice was transferred into a scent-containing cage and allowed to explore. Subsequent groups of mice were transferred at ~4–5-min intervals. After an hour of exploration, the first group of mice was tested, with vaginal moisture readings taken for each female a second time to allow comparison to her own pre-exposure baseline. Subsequent cages were tested as they reached 1 h. Experimenters (both the experimenter obtaining a sample and the individual quantifying the distance that the sample traveled along the thread) were aware of the experiment that was being conducted (i.e., they were not blinded).

#### 2.3.2. Chronic Treatments

We tested two forms of chronic exposure, one involving daily exposure to THC or cannabinoid receptor agonist CP55940 over the course of 6 days and a second testing intermittent (weekly) exposure to THC or CP55940 over the course of seven weeks. Figure 1 shows the schedule of measurements and treatments for each experiment in schematic form.

### 2.4. Statistics

Analyses were performed using Graphpad Prism (Dotmatics, Boston, MA, USA, version 10). Most experiments were analyzed using a two-tailed paired *t*-test comparing an experimental condition to the same-animal baseline. A test of population baselines in WT vs. CB1 knockout mice was tested using an unpaired *t*-test. The data from chronic treatments were analyzed using a one-way ANOVA with a Šídák post-hoc test.

### 2.5. Drugs

The phytocannabinoid THC, the chief psychoactive component of cannabis and a partial agonist at the CB1/CB2 receptors, was obtained through the NIDA Drug Supply Program. The CB1 receptor antagonist SR141716 was purchased from Biorbyt (Durham, NC, USA). The non-selective CB1/CB2 receptor agonist CP55940 was purchased from Cayman Chemical (Ann Arbor, MI, USA). These compounds were prepared for intraperitoneal injection by dissolving in a 1:1:18 mixture of ethanol, kolliphor (Sigma-Aldrich, Burlington, MA, USA), and phosphate-buffered saline (PBS). For intravaginal treatment, drugs were dissolved in 10% (2-hydroxypropyl) β cyclodextrin in PBS (pH adjusted to 7.0 with NaOH) at a concentration as noted for a given drug. Doses for drugs were chosen based on the in vivo literature (SR141716 [21], THC [22], CP55940 [23]).

## 3. Results

### 3.1. Effects of Acute Treatment with THC, CP55940, and SR141716 on Basal Vaginal Moisture

We tested whether a single acute treatment with a cannabinoid receptor agonist or antagonist would alter vaginal moisture levels relative to baseline. Unless otherwise specified, females were treated during their active phase. Females were treated with THC at 4 or 10 mg/kg, intraperitoneally (IP), without effect relative to their own baselines an hour after treatment (Figure 2A,B, THC 10 mg/kg, active phase, *n* = 8, *p*= 0.53). To test for a circadian effect, females were treated with THC during their rest phase, again seeing no effect at 4 mg/kg (Figure 2C, THC rest phase: *n* = 7, *p* = 0.92). We additionally tested whether THC might have an effect when administered locally, i.e., intravaginally (1 mM, 10 μL), but here also saw no effect of treatment (Figure 2D, THC Intravag: *n* = 9, *p* = 0.20). We next tested whether the cannabinoid receptor agonist CP55940 exerted any effects on baseline vaginal moisture, finding that it did not (Figure 2E, CP55940 IP: *n* = 10, *p* = 0.78). We additionally tested whether CP55940 might have an effect when administered intravaginally (150 μM, 10 μL), but here, too saw no effect of treatment (Figure 2F, CP55940 IVag: *n* = 13, *p* = 0.32). Lastly, we tested for an effect of the CB1 antagonist SR141716 (4 mg/kg, IP), finding that it was without effect (Figure 2G, SR141716 IP: *n* = 12, *p* = 0.11).

### 3.2. Effect of CP55940 on Stimulated Vaginal Secretion

As with other mammals, mice experience an increase in vaginal secretion at the prospect of coitus [19]. In mice this response can be stimulated variously by male bedding or urine, preputial gland extract, or the chemical messengers a/b farnesenes [19]. We tested whether the cannabinoid receptor agonist CP55940 would impair the bedding response. Mice were injected with CP55940 (0.5 mg/kg, IP) immediately before an hour-long exposure to bedding. We found that this treatment blocked the bedding response (Figure 3A, *n* = 12, *p* = 0.57 by paired t-test vs. baseline) while the response was intact in vehicle-treated mice in a same-day experiment (Figure 3B, *n* = 13, *p* = 0.0003 by paired *t*-test vs. baseline). We also tested whether local intravaginal treatment would prevent the response by treating the mice with 150 μM CP55940 (10 μL), finding that here, too, the bedding response was absent (Figure 3C*, n* = 30, *p* = 0.21 by paired *t*-test vs. baseline).

### 3.3. Effect of CB1 Receptor Block or Deletion on Stimulated Vaginal Secretion

We tested whether CB1 receptor knockout mice would have altered baseline vaginal moisture, finding that their baseline values were not different from a group of same-strain mice (Figure 4A, *n* = 30 (WT), *n* = 32 (CB1 KO); *p* = 0.43 by unpaired *t*-test). We then tested whether CB1 deletion prevents normal bedding responses. We found that CB1 knockout mice did not experience a vaginal secretory response to bedding (Figure 4B, *n* = 16, *p* = 0.82 by paired *t*-test vs. baseline). Treatment with the CB1 receptor antagonist SR141716 (4 mg/kg, IP) prevented the bedding response (Figure 4C, *n* = 24, *p* = 0.64 by paired *t*-test vs. baseline). Intravaginal treatment with SR141716 (1 mM, 10 uL) also prevented the bedding response (Figure 4D, *n* = 15, *p* = 0.74 by paired *t*-test vs. baseline).

### 3.4. Chronic THC Does Not Alter Vaginal Moisture Levels

Many cannabis users are regular consumers, and there are reports of adaptive responses to repeated exposure to cannabis [24]. We, therefore, tested for the effect of either daily or weekly administration of the phytocannabinoid THC, a partial agonist at CB1 receptors. Mice were treated with THC (4 mg/kg, IP) either daily over the course of six days or weekly over the course of 6 weeks. In each case, a final 7th injection was added to test for effects on baseline vaginal moisture. See Figure 1 for schematics of the injection timelines.

For daily treatments, we found that six days of daily THC treatment does not change the basal vaginal moisture levels (Figure 5A, Naïve BSL vs. Chronic (daily) BSL *p* = 0.77, *n* = 10). A final 7th injection of THC did not alter vaginal moisture when tested an hour post-injection (Figure 5A, Chronic (daily) BSL vs. Chronic (daily) + THC, *p* = 0.053; Naïve BSL vs. Chronic (daily) + THC, *p* = 0.89, *n* = 10).

The six weeks of weekly injections were structured similarly, comparing a pre-treatment naïve baseline with a new baseline reading one week after the 6th injection (i.e., 7 weeks after initiating injections). This chronic intermittent THC treatment did not change baseline vaginal moisture (Figure 5B, Naïve BSL vs. Chronic (weekly) BSL *p* = 0.37, *n* = 16).

Immediately after the final reading, we administered a 7th injection of THC, then tested vaginal moisture an hour after injection. Here also, the final THC injection did not change vaginal moisture levels (Figure 5B Chronic (weekly) BSL vs. Chronic (weekly) + THC *p* = 0.65, *n* = 16, Naïve BSL vs. Chronic (weekly) + THC, *p* = 0.65, *n* = 16).

### 3.5. Chronic Intermittent CP55940 Reduces Basal Vaginal Moisture and Unveils a Stimulatory CB1-Mediated Effect

Testing THC was important given that it is the chief psychoactive ingredient in cannabis, but as noted above, THC is a partial agonist at the CB1 receptor. We, therefore, again tested the full agonist, CP55940 (0.5 mg/kg, IP), with either daily or weekly treatment, as described for THC above. We found that, as with THC, six days of daily treatment with CP55940 did not alter baseline moisture levels (Figure 6A, Naïve BSL vs. Daily BSL: *p* = 0.99, 1-way ANOVA with Šídák post-hoc test, *n* = 13). A final 7th treatment with CP55940 also did not affect vaginal moisture an hour after injection (Figure 6A, Daily BSL vs. Daily + CP55940: *p* = 0.45, 1-way ANOVA with Šídák post-hoc test, *n* = 13; Naïve BSL vs. Daily + CP55940, *p* = 0.29, 1-way ANOVA with Šídák post-hoc test, *n* = 13).

While no effect was seen with daily THC and CP55940 and weekly THC, intermittent weekly treatment with CP55940 proved to have an effect. Six weeks of treatment with CP55940 (0.5 mg/kg, IP) reduced the baseline vaginal moisture levels a week after the 6th injection (Figure 6B, Naïve BSL vs. Chronic (weekly) BSL: *p* = 0.04, 1-way ANOVA with Šídák post-hoc test, *n* = 7). In addition, the final 7th treatment with CP55940 now acutely increased vaginal moisture an hour after injection (Figure 6B, Chronic (weekly) BSL vs. Chronic (weekly) + CP55940, *p* = 0.02, *n* = 7, 1-way ANOVA with Šídák post-hoc test, *n* = 7). The acute increase essentially returned values to the naïve baseline (Figure 6B, Naïve BSL vs. Chronic (weekly) + CP55940, *p* = 0.53, 1-way ANOVA with Šídák post-hoc test, *n* = 7).

Since we observed an effect of intermittent weekly CP55940, we tested for a CB1 receptor role by repeating the experiment in CB1 receptor knockouts (also on a CD1 strain background). In contrast to findings in WT mice, we did not see a drop in baseline vaginal moisture after the six-week regimen of CP55940 in CB1 knockout mice (Figure 6C, Naïve BSL vs. Chronic (weekly) BSL: *p* = 0.98, *p*-values by 1-way ANOVA with Šídák post-hoc test, *n* = 8). The final 7th injection did not result in altered vaginal moisture an hour after injection (Figure 6C, Chronic (weekly) BSL vs. Chronic (weekly) + CP55940, *p* = 0.99, naïve BSL vs. Chronic (weekly) + CP55940, *p* = 0.98, *p*-values by 1-way ANOVA with Šídák post-hoc test, *n* = 8).

## 4. Discussion

Cannabis users complain of dry eye and mouth and, in some instances, vaginal dryness, but the effects of cannabinoids on vaginal function have seen little study. We tested cannabinoid receptor activation in a recently described murine model of vaginal moisture and stimulated secretion. We found that Δ^9^-THC, the chief psychoactive ingredient in cannabis and a partial agonist at the cannabinoid CB1 receptor, has no acute effect on baseline or stimulated vaginal moisture and that the full agonist CP55940 also does not alter baseline vaginal moisture. However, in one of our chief findings, CP55940 prevents the vaginal secretory response normally induced by male pheromones. The effect of CP55940 is also seen when administered intravaginally, suggesting a local site of action. We separately tested for the effects of repeated treatments, either daily for six days or weekly for six weeks. THC again had no effects, and CP55940 had no effect after daily treatments, but in our second chief finding, six weeks of weekly CP55940 treatments lowered baseline vaginal moisture. This chronic intermittent CP55940 also unmasked a second effect, whereby acute CP55940 now increased vaginal moisture back to baseline. Our findings suggest that stimulated vaginal secretion may be regulated by CB1 receptors, while basal vaginal moisture may be impacted by chronic/intermittent CB1 activation.

### 4.1. An Effect on Stimulated but Not Basal Vaginal Moisture

In humans, vaginal moisture derives from multiple sources. Basal moisture is due to transudate diffusion across the capillary membranes in the vaginal wall [2]. However, arousal-stimulated secretions are thought to derive from glandular sources, particularly Bartholin’s glands, mucinous exocrine glands under control of the pudendal nerve [3,4]. These glands lubricate the vaginal opening and surrounding vulva. The paraurethral/Skene’s glands [5] are also implicated in vaginal lubrication [6]. There may, therefore, be a broad distinction between basal exudate-dependent moisture and stimulated exocrine-dependent secretion.

Our earlier findings that cannabinoid CB1 receptors regulate three distinct exocrine glands [9,10,11] raised the possibility that the cannabinoid signaling system also regulates vaginal-associated exocrine gland function. If cannabinoid receptors regulate the exocrine component of vaginal lubrication, one might hypothesize that cannabinoids would regulate stimulated secretion but not basal transudate-dependent vaginal moisture. Our findings for acute exposure appear to be consistent with this hypothesis, though as discussed below, the effect was only seen with a full agonist at CB1 receptors, not with THC. In addition, the situation appears to be more complex for prolonged chronic exposure, as discussed below.

### 4.2. The Mouse as a Model

As mentioned above, the use of mice as a model for vaginal function raises important questions of relevance. Mice and rats have proven to be a superlative model for many aspects of mammalian physiology, with nearly a quarter of PubMed studies making some use of them. In addition to measuring basal vaginal moisture, we make use of a pheromone-elicited vaginal stimulus to elicit vaginal secretion. How relevant is this to humans? Maintaining normative vaginal function is essential to species survival; failure to do so risks infection, injury, and the inability to thrive and reproduce. It is, therefore, likely that all mammalian species tightly regulate basal vaginal moisture as well as vaginal secretion elicited by the prospect of coitus (i.e., the ‘preparation’ response [25]). In the case of stimulated secretion, what is being tested is the *prospect of coitus*. Regardless of the cues—whether olfactory or other—every species must discern probable coitus and prepare accordingly. The preparatory machinery is then engaged. To take another example, humans do not have an ingrained fear response to feline urine, but the sympathetic machinery that is activated by fear-inducing stimuli is conserved in mice and humans. There is no reason to assume that the neuronal and physiological machinery implementing this response to the prospect of coitus is somehow profoundly and fundamentally different in humans vs. mice.

It is true that mice are very small relative to humans and also nocturnal; the recently described model of murine vaginal secretion remains to be fully ‘road-tested’. In addition, limited anatomical work has been performed to characterize potential sources of vaginal secretion in mice. The Skene’s glands appear to have an analog in rodents (e.g., [26]) but it is unclear whether female mice have a Bartholin’s gland. Instead, in this location, rodents have what is referred to as a clitoral gland that is proposed to be unique to rodents and to act in scent marking [27,28]. This sizeable gland is well-positioned to also assist in lubrication of the vaginal opening and vulva in preparation for intercourse, but the exocrine component of stimulated vaginal secretion in rodents remains to be fully characterized.

### 4.3. Potential Mechanisms

Intravaginal treatments suggest a local site of action. In previous studies of CB1 regulation of the lacrimal, submandibular, and parotid glands, we have found a similar architecture of the cannabinoid signaling system, with CB1 receptors positioned presynaptically on cholinergic parasympathetic inputs. Our hypothesized mechanism of action is that CB1 receptors inhibit autonomic cholinergic inputs, and a similar arrangement may hold for peri-vaginal exocrine glands, though this remains to be examined. It should also be noted that we found a notable exception in female lacrimal glands. In addition to CB1 receptors on cholinergic inputs, female lacrimal glands express a second population of CB1 receptors in myoepithelial cells that may contribute to circadian regulation of tearing [12].

Chronic treatments produced very different results that depended both on the agonists employed and the frequency/duration of exposure. Daily treatments were without effect, but chronic intermittent CP55940 reduced basal vaginal moisture, likely indicating that some tolerance develops in response to chronic intermittent CB1 activation. This may be due to activation-induced downregulation of receptors or changes in downstream signaling. The chronic intermittent schedule also revealed an unexpected potentiating effect of CP55940. The increase did not exceed the original naïve baseline but may point to a second site of action for CP55940 that is unmasked or induced by the chronic treatment.

### 4.4. THC, CP55940, and the Potential Effects of Cannabis

Given that the study was prompted in part by informal cannabis user complaints of dry mouth, eye, and vaginal dryness, it is notable that we did not see an effect of THC itself. We chose 4 mg/kg as a dose that is both commonly used and found to be effective in mice, but THC is a partial agonist (reviewed in [29,30]). It was with the full agonist CP55940 that we observed several pronounced effects on vaginal lubrication. In addition to the caveats of using an animal model discussed above, there is also little available research on humans. One survey that examined the sexual satisfaction and function of female cannabis users included a question about vaginal lubrication, reporting no significant effect [8]. It is possible that the effects on vaginal moisture in cannabis smokers are dose-dependent or that there are subpopulations that are more sensitive to these effects of cannabis. Our evidence pointing to local peri-vaginal effects also raises the possibility that THC-infused products, such as vaginal creams and condoms, may have unexpected side effects, particularly at high local doses.

### 4.5. Ruling out the Role of Olfactory Cannabinoid Receptors

A recent study found that an intact olfactory CB1 receptor signaling system was necessary to communicate stress between mice [31]. We have similarly found that CB1 receptors are necessary for the perception of some non-pheromonal scents [11]. This raised the possibility that the absence of a stimulated secretory response in CB1 knockout mice was due to a requirement for olfactory CB1 receptors in the detection of pheromonal olfactory cues. The positive effects with intravaginal CP55940 and the block with intravaginal SR141716 support the idea that the effects on stimulated vaginal secretion are due to local vaginal or peri-vaginal action.

## 5. Conclusions

Changes in the legal status of cannabis both in the US and abroad have been accompanied by an enthusiastic embrace of all things cannabis both by consumers and commercial entities. Almost nothing is known about the effects of cannabinoids on vaginal health and function. This has not prevented companies from stepping into the void with phytocannabinoid-infused products accompanied by claims of beneficial effects. To shed light on the subject, we have developed a mouse model of vaginal moisture and stimulated secretion and, using this, find evidence for complex regulation of vaginal moisture. In the current studies, THC itself was without effect, and a single exposure to cannabinoid receptor agonists did not alter basal vaginal moisture. However, acute treatment with the full CB1 agonist CP55940 did prevent pheromone-stimulated vaginal secretion. Chronic intermittent CB1 activation reduced basal vaginal moisture. It remains to be determined whether these findings in a laboratory mouse model will translate to other animals, but the functions of the vagina, including the tight regulation of the intravaginal environment and secretory response to the prospect of coitus, are shared by all species and may be highly conserved.

## Figures and Tables

**Figure 1 biomolecules-15-00472-f001:**
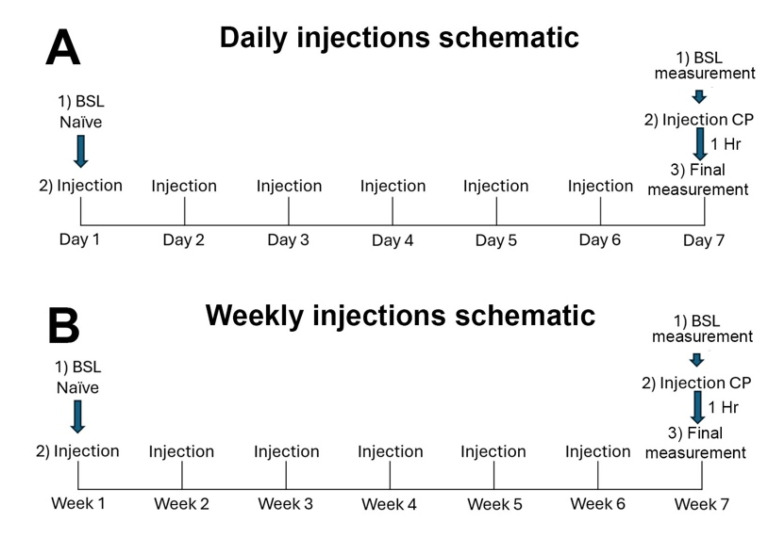
Schematic of injections for daily and weekly treatments. (**A**) For repeated daily treatments, a first vaginal moisture measurement is taken before the first injection (“BSL Naïve”). A day after the 6th injection, a measurement is taken, “BSL measurement”, followed by a final 7th injection. A reading “Final measurement” is taken after an hour. (**B**) As above, but with weekly injections.

**Figure 2 biomolecules-15-00472-f002:**
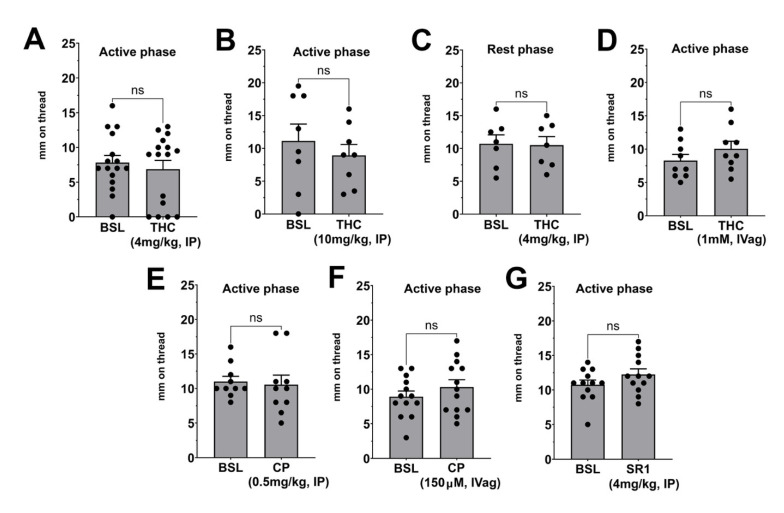
Effects of acute treatment with THC, CP55940, and SR141716 on basal vaginal moisture**. A**–**C**) Effects of THC on basal vaginal moisture were tested during the active phase at 4 mg/kg (**A**) and 10 mg/kg (**B**) and also in the rest phase at 4 mg/kg (**C**). (**D**) Intravaginally applied THC (1 mM, 10 μL) also had no effect. (**E**,**F**) Effects of CP55940 were tested when applied (**E**) intraperitoneally (0.5 mg/kg, IP) or (**F**) intravaginally (150 μM, 10 μL). (**G**) Effects of CB1 antagonist SR141716 (4 mg/kg, IP) were tested. ns, not significant by paired *t*-test vs. baseline.

**Figure 3 biomolecules-15-00472-f003:**
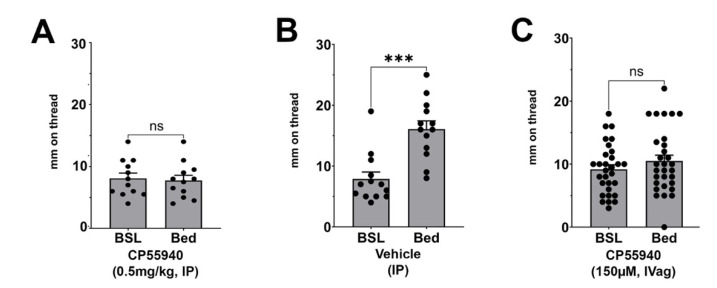
CB1 agonist effect on bedding-stimulated vaginal secretory response. (**A**) The bedding-stimulated secretory response is absent in mice pretreated intraperitoneally with CB1 agonist CP55940 (0.5 mg/kg); (**B**) Response to bedding is present in vehicle-treated mice. (**C**) Bedding response is also absent when CP55940 treatment is intravaginal (150 μM, 10 μL). ns, not significant, ***, *p* < 0.001 by paired *t*-test vs. same-animal baseline.

**Figure 4 biomolecules-15-00472-f004:**
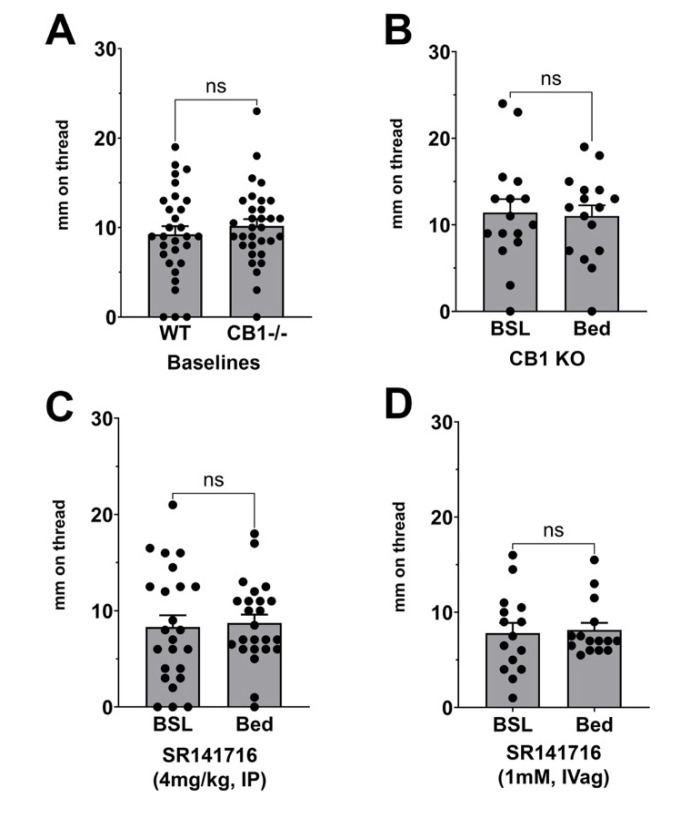
Effect of CB1 receptor block or deletion on stimulated vaginal secretion. (**A**) Baseline vaginal moisture is the same in CB1 knockout (KO) vs. same-strain controls. (**B**) Stimulated secretion is absent in CB1 knockout mice. (**C**) No response to bedding is seen in mice treated with SR141716 intraperitoneally (4 mg/kg) or intravaginally ((**D**) 1 mM, 10 μL). ns, not significant by unpaired (**A**) or paired (**B**–**D**) *t*-test vs. same-animal baseline.

**Figure 5 biomolecules-15-00472-f005:**
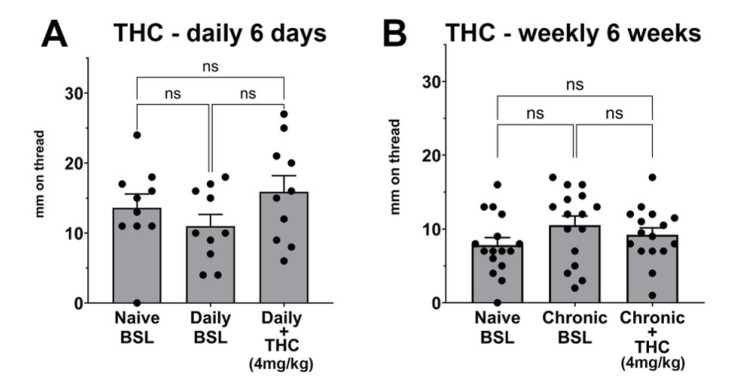
Chronic THC does not alter vaginal moisture levels. (**A**) Mice were treated with THC (4 mg/kg, IP) daily for six days. Naïve BSL reading from before first injection; Chronic BSL one day after 6th injection; Chronic + THC: 1 h after 7th injection. (**B**) Mice were treated with THC (4 mg/kg, IP) weekly for six weeks. Naïve BSL reading from before first injection; Chronic BSL one week after 6th injection; Chronic + THC: 1 h after 7th injection. ns, not significant by 1-way ANOVA with Šídák post-hoc test.

**Figure 6 biomolecules-15-00472-f006:**
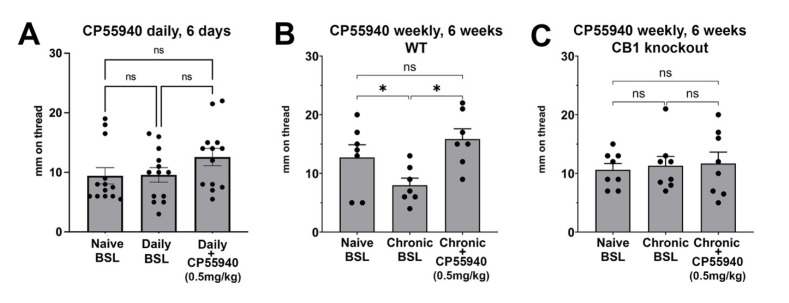
Effects of chronic intermittent CP55940 on vaginal moisture. (**A**) Mice were treated with CP55940 (0.5 mg/kg, IP) daily for six days. Naïve BSL reading from before first injection; Chronic BSL one day after 6th injection; Chronic + CP55940: 1 h after 7th injection. (**B**,**C**) CD1 strain (**B**) and CB1 KO on CD1 strain background (**C**) were treated with CP55940 (0.5 mg/kg, IP) weekly for six weeks. Naïve baseline (BSL) reading from before first injection; Chronic BSL one week after 6th injection; Chronic + CP: 1 h after 7th injection. ns, not significant; * *p* < 0.05; 1-way ANOVA with Šídák post-hoc test.

## Data Availability

The data underlying this article will be shared on request to the corresponding author.

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
