# Peer review of "Cannabinoid Regulation of Murine Vaginal Secretion"

_biomolecules, 2025, doi:10.3390/biom15040472_

Round 1

Reviewer 1 Report

Comments and Suggestions for Authors

In their manuscript, the authors show that cannabinoid signaling as involved in the regulation of vaginal secretion. The manuscript is interesting, and may have the potential to attract the attention of a wide-array of readers. However, I think that there are critical errors in the experimental design that question the interpretation of the findings.

Major points:

1.       Figure 2. The authors claim that “The bedding-stimulated secretory response is absent in mice pretreated with CB1 agonist CP55940…”. However, the do not show the normal bedding response. Considering the huge variability of the measurement (mean of BSL moisture level on panel A is ca. half of the one shown on panel B), they should have included a vehicle-treated control group as well, and they should have compared the effect of CP to the effect of the vehicle. Please repeat the experiment.

2.       Line 240 “A follow-on treatment with the CB1 receptor antagonist SR141716 did not reverse this effect.” Why should it? On one hand, because we do not know the time course of the vaginal secretion, we simply cannot compare CP and CP+SR, or THC and THC+SR treatments to each other. On the other hand, you should have treated the mice with the antagonist before the application of the agonists. Indeed, in case of Figures 3-4, the authors measured the effects of THC and CP. Then, they treated the mice with the CB1 antagonist, and, one hour later, they repeated the measurement. This experimental design is inappropriate to investigate the role of CB1 in mediating the effects of CP or THC. The authors should have created an independent group animals of which should have received CB1 antagonist prior to every THC/CP treatment. Please repeat the experiments by using an appropriate experimental design.

3.       In order to investigate the role of CB1 in the process, please compare both the unstimulated baseline vaginal secretion as well as the bedding-stimulated vaginal secretion of the CB1-/- mice to wild-type littermates.

4.       Figure 5. The effect of CBD on bedding-stimulated secretion should also be tested.

Minor points:

1.       There are several typos (e.g., unnecessary space in line 40, multiple commas in line 126, missing spaces between data and their units, “uM” and “ul” instead of “μM” and “μl”, etc.) in the text. Please correct them all.

2.       Please spell “Sjogren” correctly as “Sjögren”.

3.       Please educate the readers by explaining your previous findings in more details. Specifically, when talking about “a notable sex-dependent exception” in the regulatory role of cannabinoid signaling in various glands, please tell the readers what that difference was.

4.       Lines 57-58 “A second major component of cannabis, cannabidiol (CBD), has been the subject of intensive study and is approved as a therapy for some forms of epilepsy.” This is true; but you should mention that THC is also in use in the clinical practice alone as well as in combination with CBD.

5.       Lines 66-69 “These represent high local doses and, given that the lipophilic phytocannabinoids can be readily absorbed through the vaginal wall, may also produce systemic effects. If they do alter vaginal moisture they may in turn impact local flora or vaginal pH, with attendant risk of vaginal disorders.

a.       This is true. However, it is noteworthy that, together with many other phytocannabinoids, CBD has already been shown to exert direct anti-microbial effects (see e.g., PMID: 18681481) as well. In fact, there are ongoing clinical trials (https://botanixpharma.com/products-pipeline/) aiming at exploring the potential of CBD against certain bacteria. Thus, such high local CBD doses most likely have a direct impact on the vaginal flora. Please mention this possibility in the text.

b.      Please add an appropriate reference to support your statement on the systemic absorption, and provide more data (if available) about the bioavailability of phytocannabinoids following vaginal administration.

6.       Methods:

a.       Were the animals kept under SPF conditions? Please specify.

b.      Description of the use of the animals (lines 95-120) is somewhat confusing and vague. Please insert here the Supplementary Figure. It will help explaining the study design. Moreover, please add the doses wherever they are missing.

c.       Please specify the version of GraphPad Prism.

d.      Please spell “Sidak” correctly as “Šídák”.

e.       Why did you assume Gaussian distribution? GraphPad can test this for you very easily. Please check the distribution of the data, and in case of non-Gaussian distribution, use a non-parametric test.

7.       Results:

a.       Use color code on the Figures to identify the animals so that the reader can follow the behavior of the same animal across the groups.

b.      Figure 1: The doses are not “mg”, but “mg/kg”, right? Likewise, please add the volume in which the 10 mM solution of CBD was administered on Figure 5B.

8.       Line 270 “Tearing and salivation are wholly, and vaginal secretion partly, exocrine in nature”. It is most definitely not endocrine… so what do you mean by “partly” in this context?

9. When discussing the effects of chronic CB1 activation, please also mention the possibility of activation-induced down-regulation, and development of pharmacological tolerance.

Author Response

We thank the reviewers for their close attention to the manuscript, and also for the rapidity of their review.   We also thank the reviewers for the constructive nature of the comments.  We have attended closely to these comments and suggestions and have conducted several additional experiments and substantially modified the manuscript.  We consider the manuscript improved by these changes. 

We have listed the comments and our replies below.    We have also substantially revised the structure of the manuscript to include requested results from knockout mice as well as follow-on experiments with the CB1 receptor antagonist SR141716.   The discussion has been substantially reorganized, including headings.  The CBD findings will now be the subject of a separate follow-on study.   In our responses below, we have modified the reviewer comments to facilitate a point-by-point response.  In some cases, we have omitted favorable comments by reviewers to focus on areas that needed improvement/attention.

Separately, the editor noted that our percentage of self-citations was a reason for concern.   Seven articles from our research group did not seem unreasonable to us, especially given that the current work is essentially a new field of study using a model recently developed by our group limits the number of potential other work that we might have cited.   We have worked in the cannabinoid field for ~25 years and our work includes review articles that are apropos to the subject matter.   That said, we applaud the efforts of the journal to set and enforce journal standards and have replaced our two review articles, taking the number of citations to 5 (out of 30).   If necessary we can individually justify the remaining references.

Reviewer 1:

I think that there are critical errors in the experimental design that question the interpretation of the findings.

Major points:

  1. Figure 2. The authors claim that “The bedding-stimulated secretory response is absent in mice pretreated with CB1 agonist CP55940…”. However, the do not show the normal bedding response. Considering the huge variability of the measurement (mean of BSL moisture level on panel A is ca. half of the one shown on panel B), they should have included a vehicle-treated control group as well, and they should have compared the effect of CP to the effect of the vehicle. Please repeat the experiment.

A:  We have repeated the CP55940 experiment in panel A, including a vehicle-treated group in parallel. 

Regarding the statistical analyses, there is some variability in baseline vaginal moisture.  This is why we compare mice to their own baselines.  Given the baseline differences and the fact that we are able to test mice against their own baselines, we do not believe that a statistical comparison between vehicle and CP treated groups is beneficial here.

  1. Line 240 “A follow-on treatment with the CB1 receptor antagonist SR141716 did not reverse this effect.” Why should it? On one hand, because we do not know the time course of the vaginal secretion, we simply cannot compare CP and CP+SR, or THC and THC+SR treatments to each other. On the other hand, you should have treated the mice with the antagonist before the application of the agonists. Indeed, in case of Figures 3-4, the authors measured the effects of THC and CP. Then, they treated the mice with the CB1 antagonist, and, one hour later, they repeated the measurement. This experimental design is inappropriate to investigate the role of CB1 in mediating the effects of CP or THC. The authors should have created an independent group animals of which should have received CB1 antagonist prior to every THC/CP treatment. Please repeat the experiments by using an appropriate experimental design.

Adding in a final treatment with SR1 was an opportunity to learn whether the circuit/system has adapted over the course of days or weeks of treatment.  This is especially relevant in a situation where there may be more than one site of action, with different desensitization. We found the differential effects to be instructive.   The experiment proposed by the reviewer (i.e. pretreat the animals with SR1) would only serve as the pharmacological parallel to CB1 knockout experiments.

There is (especially now) little appetite for funding these sorts of studies, and as it is this was pilot data.  Rather than repeat a six-week study, we have removed the final-injection SR1 data from each figure and re-analyzed the remaining data accordingly.

  1. In order to investigate the role of CB1 in the process, please compare both the unstimulated baseline vaginal secretion as well as the bedding-stimulated vaginal secretion of the CB1-/- mice to wild-type littermates.

We have added in the responses to bedding in CB1 KO mice as well as the bedding response in CP55940-treated mice.

We have also added a comparison of CB1 KO baselines to mice on the same background strain (not littermates).  Littermates are more important for the study of behavioral responses that are often quite variable and for which parental influence can play a key role.  Here we are measuring a robust physiological response that is highly conserved across species and that we have demonstrated in two mouse strains (C57BL/6 and CD1).

  1. Figure 5. The effect of CBD on bedding-stimulated secretion should also be tested.

Our ongoing experiments suggest that the CBD effect would work better as a standalone story that explores the mechanism of action in detail.  We have removed this figure and updated the manuscript accordingly.

Minor points:

  1. There are several typos (e.g., unnecessary space in line 40, multiple commas in line 126, missing spaces between data and their units, “uM” and “ul” instead of “μM” and “μl”, etc.) in the text. Please correct them all.

We have corrected these.

  1. Please spell “Sjogren” correctly as “Sjögren”.

The o has been changed to ö.

  1. Please educate the readers by explaining your previous findings in more details. Specifically, when talking about “a notable sex-dependent exception” in the regulatory role of cannabinoid signaling in various glands, please tell the readers what that difference was.

We have added details on this in the introduction and discussion.

  1. Lines 57-58 “A second major component of cannabis, cannabidiol (CBD), has been the subject of intensive study and is approved as a therapy for some forms of epilepsy.” This is true; but you should mention that THC is also in use in the clinical practice alone as well as in combination with CBD.

We have added several approved therapeutic uses for THC.

  1. Lines 66-69 “These represent high local doses and, given that the lipophilic phytocannabinoids can be readily absorbed through the vaginal wall, may also produce systemic effects. If they do alter vaginal moisture they may in turn impact local flora or vaginal pH, with attendant risk of vaginal disorders.
  2. This is true. However, it is noteworthy that, together with many other phytocannabinoids, CBD has already been shown to exert direct anti-microbial effects (see e.g., PMID: 18681481) as well. In fact, there are ongoing clinical trials (https://botanixpharma.com/products-pipeline/) aiming at exploring the potential of CBD against certain bacteria. Thus, such high local CBD doses most likely have a direct impact on the vaginal flora. Please mention this possibility in the text.
  3. Please add an appropriate reference to support your statement on the systemic absorption, and provide more data (if available) about the bioavailability of phytocannabinoids following vaginal administration.

As noted, the CBD section has been excised from the manuscript.

  1. Methods:
  2. Were the animals kept under SPF conditions? Please specify.

The animals were maintained in a SPF facility.  This is now noted in the methods.

  1. Description of the use of the animals (lines 95-120) is somewhat confusing and vague. Please insert here the Supplementary Figure. It will help explaining the study design. Moreover, please add the doses wherever they are missing.

We have brought the Supplementary figure into the main body of text as a new Figure 1 along with an accompanying explanatory paragraph.  We have also added in doses. 

  1. Please specify the version of GraphPad Prism.

Done.

  1. Please spell “Sidak” correctly as “Šídák”.

Done

  1. Why did you assume Gaussian distribution? GraphPad can test this for you very easily. Please check the distribution of the data, and in case of non-Gaussian distribution, use a non-parametric test.

This is residual language inserted in response to a reviewer query on our first manuscript (Murataeva et al., 2024) where we confirmed Gaussian distribution in a sample data set.   We have removed the sentence.

  1. Results:
  2. Use color code on the Figures to identify the animals so that the reader can follow the behavior of the same animal across the groups.

If we use a color code in this way it will need to be applied throughout the manuscript.  We tried this but do not believe that this will prove to be effective/useful for readers, especially where there are a large number of data points.  As an alternative we also tried before-after plotting but the result, especially for the chronic treatments was quite messy.   We advise against these. 

  1. Figure 1: The doses are not “mg”, but “mg/kg”, right?

This has been corrected

Likewise, please add the volume in which the 10 mM solution of CBD was administered on Figure 5B.

This figure has been removed.

  1. Line 270 “Tearing and salivation are wholly, and vaginal secretion partly, exocrine in nature”. It is most definitely not endocrine… so what do you mean by “partly” in this context?

            A: This is discussed in the second paragraph and involves transudation through the vaginal walls.  We have moved that sentence into the second paragraph to integrate it with the subsequent discussion.

  1. When discussing the effects of chronic CB1 activation, please also mention the possibility of activation-induced down-regulation, and development of pharmacological tolerance.

A: We have added a sentence about potential mechanisms that might underlie the development of tolerance.

Reviewer 2 Report

Comments and Suggestions for Authors

1.      Please include references for the doses used for the tested drugs.

2.      Why did the authors use the CD1 strain in this model, is this strain more susceptible to induce this model than other strains?

3.      Illustrating a figure to show the treatment duration clearly is recommended.

Author Response

We thank the reviewers for their close attention to the manuscript, and also for the rapidity of their review.   We also thank the reviewers for the constructive nature of the comments.  We have attended closely to these comments and suggestions and have conducted several additional experiments and substantially modified the manuscript.  We consider the manuscript improved by these changes. 

We have listed the comments and our replies below.    We have also substantially revised the structure of the manuscript to include requested results from knockout mice as well as follow-on experiments with the CB1 receptor antagonist SR141716.   The discussion has been substantially reorganized, including headings.  The CBD findings will now be the subject of a separate follow-on study.   In our responses below, we have modified the reviewer comments to facilitate a point-by-point response.  In some cases, we have omitted favorable comments by reviewers to focus on areas that needed improvement/attention.

Separately, the editor noted that our percentage of self-citations was a reason for concern.   Seven articles from our research group did not seem unreasonable to us, especially given that the current work is essentially a new field of study using a model recently developed by our group limits the number of potential other work that we might have cited.   We have worked in the cannabinoid field for ~25 years and our work includes review articles that are apropos to the subject matter.   That said, we applaud the efforts of the journal to set and enforce journal standards and have replaced our two review articles, taking the number of citations to 5 (out of 30).   If necessary we can individually justify the remaining references.

Reviewer 2: 

  1. Please include references for the doses used for the tested drugs.

A: We have added references under drugs.

  1. Why did the authors use the CD1 strain in this model, is this strain more susceptible to induce this model than other strains?

A: We started with C57Bl/6 mice (which is why we had some extra ‘free’ data), but learned in parallel experiments that CB1 receptors play a role in circadian regulation of stimulated vaginal secretion (Murataeva et al., 2024) and tearing (Murataeva et al., 2024b).  C57BL/6 mice are melatonin deficient and so do not always have normal circadian phenotypes.  CD1 strain mice a clear circadian regulation of stimulated vaginal responses and also of tearing and are additionally good breeders with large litters.  We have added some details relating to strain choice to the methods.

  1. Illustrating a figure to show the treatment duration clearly is recommended.

Reviewer 1 concurs.  We have moved the schematic from supplemental to the methods section.

Reviewer 3 Report

Comments and Suggestions for Authors

The paper entitled “Cannabinoid regulation of murine vaginal secretion” evaluates the effect of cannabinoid receptor type 1 agonists and antagonists on sexual arousal-related vaginal secretion in female mice using a novel detection method. The research topic is groundbreaking since the study of female sexual responses and vaginal function is scarce. The results indicate that the administration (acute and chronic) of a full agonist of CB1 receptors can modify vaginal secretion. Also, the systemic administration of the phytocannabinoid CBD can increase the secretion. The implications of these findings open new horizons for the study of the effects of cannabinoids.

The manuscript is suitable; however, I have some comments that can improve the manuscript's clarity. The method and result sections need to be improved. My comments are as follows:

In the abstract, the authors mention, “Using this model, we tested the regulation of vaginal moisture by cannabinoid receptors.” However, they tested only the agonist and antagonist of CB1 receptors, as well as another phytocannabinoid. So, the first statement is not adequate.

The Introduction section is adequate for understanding the scientific problem statement.

The Method section can be improved.

The authors did not include an “Experimental design” subtitle; instead, they enlisted the groups of treatments and drugs (lines 96 - 118). However, this is hard to understand; to this point, the drugs employed are not even mentioned yet, and the use of abbreviations demands to the reader to look up to the compounds in the following sections. Also, the document writing in that section (lines 92 - 96) is hard to read; the use of “:” is not followed by a list, and the information is replicated. In general, the information under “Study Animals” subtitle is not clear.

It is valid to use more than one mouse strain; however, it is not clear when the different stains are used for each experiment. Also, the is no mention of the CB1R knockout mice in the animal section.

The intravaginal treatment is not clearly described to ensure replicability. Since the sample was taken from the administration site, the authors can explain how they designed this experiment in terms of the time between administration and evaluation.

Also, the mechanism of action of the drugs used can be described in the “Drugs” section since it is specified late in the manuscript (Results section).

The abbreviation of “reverse light cycle” (RLC) is hard to find in the manuscript (first appearance on line 97), making it confusing for the reader. It is even more confusing to use the “rest phase” (SLC) abbreviation that is mentioned in a figure legend (line 185). The description and logic for considering the circadian rhythms and, therefore, artificially setting up a cycle is understandable. Still, the authors can make it simple to present the specification of it in the Results section.

Why THC is not tested intravaginally, but CP55940 is?

The use of vehicle-treated animals is missing; the authors must justify it. I understand the comparisons with baseline, but using a control group in some experiments can augment the strength of the result, mainly when the authors conclude that the: “treatment blocked the bedding response” for instance.

The Results section can be improved in terms of clarity.

Figure legend from Figure 1, did not include the descriptive statistic considered or the test (even if it is not significant).

The first subtitle states: “3.1 Effects of acute treatment with THC and CP55940 on basal vaginal moisture” but the antagonist was also tested.

The second subtitle is “3.2. Effect of cannabinoid receptor agonists on stimulated vaginal secretion”. But the only compound tested was CP55940.

The authors mentioned that THC is a partial agonist of CB1R (line 234), so they tested the effect of a full agonist (CP55940) but then mentioned that THC is a CB1R agonist (line 205).

There is no apparent justification for using a specific drug for chronic or subchronic treatments. For example, why is CP55940, which is the only treatment that modifies the variable, not tested in the daily 6-day protocol?

Overall, the result section is hard to read, and there is no easy way to come back to the methods and understand them in terms of congruence. Maybe reorganizing and improving the subtitle can make the results easier to follow.

Minor comments

Some typos need to be carefully revised, for example:

1.      An extra “,” is found after the word Briefly (line 126)

2.      The doses are specified on some “Y” axes of the graphs but not others. I suggest making it consistent.

3.      Double point (line 188)

Author Response

We thank the reviewers for their close attention to the manuscript, and also for the rapidity of their review.   We also thank the reviewers for the constructive nature of the comments.  We have attended closely to these comments and suggestions and have conducted several additional experiments and substantially modified the manuscript.  We consider the manuscript improved by these changes. 

We have listed the comments and our replies below.    We have also substantially revised the structure of the manuscript to include requested results from knockout mice as well as follow-on experiments with the CB1 receptor antagonist SR141716.   The discussion has been substantially reorganized, including headings.  The CBD findings will now be the subject of a separate follow-on study.   In our responses below, we have modified the reviewer comments to facilitate a point-by-point response.  In some cases, we have omitted favorable comments by reviewers to focus on areas that needed improvement/attention.

Separately, the editor noted that our percentage of self-citations was a reason for concern.   Seven articles from our research group did not seem unreasonable to us, especially given that the current work is essentially a new field of study using a model recently developed by our group limits the number of potential other work that we might have cited.   We have worked in the cannabinoid field for ~25 years and our work includes review articles that are apropos to the subject matter.   That said, we applaud the efforts of the journal to set and enforce journal standards and have replaced our two review articles, taking the number of citations to 5 (out of 30).   If necessary we can individually justify the remaining references.

Reviewer 3:

In the abstract, the authors mention, “Using this model, we tested the regulation of vaginal moisture by cannabinoid receptors.” However, they tested only the agonist and antagonist of CB1 receptors, as well as another phytocannabinoid. So, the first statement is not adequate.

We have corrected this.

The Method section can be improved.

The authors did not include an “Experimental design” subtitle; instead, they enlisted the groups of treatments and drugs (lines 96 - 118). However, this is hard to understand; to this point, the drugs employed are not even mentioned yet, and the use of abbreviations demands to the reader to look up to the compounds in the following sections. Also, the document writing in that section (lines 92 - 96) is hard to read; the use of “:” is not followed by a list, and the information is replicated. In general, the information under “Study Animals” subtitle is not clear.

We have revised this section.

It is valid to use more than one mouse strain; however, it is not clear when the different stains are used for each experiment.

We understand that the C57 data for the chronic CP experiment seems out of place.  It was ‘free data’ but doesn’t add much, especially without the final SR141716 treatment, so we are leaving it out and have accordingly removed most mention of that mouse strain except where providing a rationale for the use of CD1 strain mice.

The is no mention of the CB1R knockout mice in the animal section.

Information about the CB1 knockouts and the CD1 strain have been added.

The intravaginal treatment is not clearly described to ensure replicability. Since the sample was taken from the administration site, the authors can explain how they designed this experiment in terms of the time between administration and evaluation.

We have added a paragraph to the methods that explicitly maps out how this experiment was done.

Also, the mechanism of action of the drugs used can be described in the “Drugs” section since it is specified late in the manuscript (Results section).

We have added information about the mechanism of action of drugs to this section.  

The abbreviation of “reverse light cycle” (RLC) is hard to find in the manuscript (first appearance on line 97), making it confusing for the reader. It is even more confusing to use the “rest phase” (SLC) abbreviation that is mentioned in a figure legend (line 185). The description and logic for considering the circadian rhythms and, therefore, artificially setting up a cycle is understandable. Still, the authors can make it simple to present the specification of it in the Results section.

We agree that this could be clearer and more consistent.   We have made the terminology more uniform and cleared by using active vs. rest phase.

Why THC is not tested intravaginally, but CP55940 is?

A: We have added a panel to Figure 1 (now Fig 2) with this data.

The use of vehicle-treated animals is missing; the authors must justify it. I understand the comparisons with baseline, but using a control group in some experiments can augment the strength of the result, mainly when the authors conclude that the: “treatment blocked the bedding response” for instance.

A: Reviewer 1 also requested a vehicle control.  We repeated the bedding experiment with a same-day vehicle-treated group and agree that this makes the non-effect clearer.

The Results section can be improved in terms of clarity.

Figure legend from Figure 1, did not include the descriptive statistic considered or the test (even if it is not significant).

This detail has been added to the figure legend.

The first subtitle states: “3.1 Effects of acute treatment with THC and CP55940 on basal vaginal moisture” but the antagonist was also tested.

This subtitle has been updated.

The second subtitle is “3.2. Effect of cannabinoid receptor agonists on stimulated vaginal secretion”. But the only compound tested was CP55940.

This has been updated.

The authors mentioned that THC is a partial agonist of CB1R (line 234), so they tested the effect of a full agonist (CP55940) but then mentioned that THC is a CB1R agonist (line 205).

The text has been updated.

There is no apparent justification for using a specific drug for chronic or subchronic treatments. For example, why is CP55940, which is the only treatment that modifies the variable, not tested in the daily 6-day protocol?

A: We agree that this could be instructive, and have repeated the six-day experiment with CP55940.  We find that there is no effect. 

Overall, the result section is hard to read, and there is no easy way to come back to the methods and understand them in terms of congruence. Maybe reorganizing and improving the subtitle can make the results easier to follow.

A: We have revised the results and hope that they are now easier to follow.   We have also substantially reorganized the discussion.

Minor comments

Some typos need to be carefully revised, for example:

  1. An extra “,” is found after the word Briefly (line 126)
  2. The doses are specified on some “Y” axes of the graphs but not others. I suggest making it consistent.
  3. Double point (line 188)

We have given the manuscript an additional scrubbing to correct typos etc.  

Round 2

Reviewer 1 Report

Comments and Suggestions for Authors

-

Reviewer 3 Report

Comments and Suggestions for Authors

The paper has improved substantially. I do not have any more suggestions.